# Functional evolution of Lepidoptera olfactory receptors revealed by deorphanization of a moth repertoire

Arthur de Fouchier[1,*,†], William B. Walker III[2,*], Nicolas Montagné[3], Claudia Steiner[2,3], Muhammad Binyameen[2,4,†], Fredrik Schlyter[2,†], Thomas Chertemps[3], Annick Maria[3], Marie-Christine François[1], Christelle Monsempes[1], Peter Anderson[2], Bill S. Hansson[5], Mattias C. Larsson[2] & Emmanuelle Jacquin-Joly[1]

Insects detect their hosts or mates primarily through olfaction, and olfactory receptors (ORs) are at the core of odorant detection. Each species has evolved a unique repertoire of ORs whose functional properties are expected to meet its ecological needs, though little is known about the molecular basis of olfaction outside Diptera. Here we report a pioneer functional analysis of a large array of ORs in a lepidopteran, the herbivorous pest *Spodoptera littoralis*. We demonstrate that most ORs are narrowly tuned to ubiquitous plant volatiles at low, relevant odorant titres. Our phylogenetic analysis highlights a basic conservation of function within the receptor repertoire of Lepidoptera, across the expansive evolutionary radiation of different major clades. Our study provides a reference for further studies of olfactory mechanisms in Lepidoptera, a historically crucial insect order in olfactory research.

[1] INRA, Institute of Ecology & Environmental Sciences of Paris, Department of Sensory Ecology, Route de Saint-Cyr, 78026 Versailles Cedex, France. [2] Department of Plant Protection Biology, Swedish University of Agricultural Sciences, Sundsvägen 14, 230 53 Alnarp, Sweden. [3] Sorbonne Universités—UPMC University Paris 06, Institute of Ecology & Environmental Sciences of Paris, Department of Sensory Ecology, 7 quai Saint Bernard, 75252 Paris Cedex 05, France. [4] Chemical Ecology Laboratory, Department of Entomology, Bahauddin Zakariya University, Multan 60800, Pakistan. [5] Department of Evolutionary Neuroethology, Max Planck Institute for Chemical Ecology, 07745 Jena, Germany. * These authors contributed equally to this work. † Present addresses: Institute for Biodiversity and Ecosystems Dynamics, University of Amsterdam, 1098 XH Amsterdam, The Netherlands (A.d.F.); Faculty of Forestry and Wood Sciences, Czech University of Life Sciences, Kamýcká 1176, Prague 6, Suchdol 165 21, Czech Republic (M.B. and F.S.). Correspondence and requests for materials should be addressed to M.C.L. (email: mattias.larsson@slu.se) or to E.J.-J. (email: emmanuelle.joly@inra.fr).

Ecological interactions between insects and plants are regulated, to a great extent, by the emission of volatile organic compounds by flowers, fruits and leaves. The olfactory perception of these airborne cues drives vital behaviours, such as foraging and oviposition site selection. Insects detect odorants through olfactory sensory neurons (OSNs) housed within sensory hairs—the olfactory sensilla—that cover the surface of antennae and maxillary palps[1]. The watershed event that allowed deciphering the molecular basis of odorant detection in insects was the discovery of olfactory receptor (OR) genes in the model Drosophila melanogaster[2–4]. These genes encode transmembrane proteins that bind chemical stimuli at the surface of OSNs and mediate signal transduction[1]. Each OSN usually expresses a highly conserved co-receptor named Orco[5,6] together with one OR that is responsible for the odour response spectrum of the OSN. Another family of chemosensory receptors expressed in a smaller population of OSNs, named variant ionotropic receptors (IRs), is tuned to complementary chemical classes of odorants[7]. The olfactory capacities of an insect thus depend to a great extent on the repertoire of expressed OR genes and on the functional properties of OR proteins, notably their sensitivity and the breadth of their response spectra.

Many insect OR repertoires have now been identified and unique lineage-specific expansions of OR clades have been observed in the different insect orders[1]. This observation suggests that in each order, ORs have followed different evolutionary trajectories as insects adapted to new ecological niches. Such adaptation has been investigated in Diptera, thanks to the functional characterization of the OR repertoires from D. melanogaster[8–12] and Anopheles gambiae, the primary malaria vector mosquito[13,14]; these pioneering works notably demonstrated that the OR repertoires of these organisms are specialized for the detection of ecologically relevant chemicals. However, such specialization remains to be investigated in other insect orders with distinct evolutionary histories and food preferences, as no large OR repertoire has yet been functionally characterized outside Diptera. For instance, only a few OR-ligand couples have been identified in herbivorous insects apart from moth sex pheromone receptors[15]. Chemoecological studies of herbivorous insects from different orders have led to the hypothesis that they generally recognize host and non-host plants by detecting specific combinations of a relatively limited number of ubiquitous aliphatics, aromatics and terpenes, rather than taxon-specific volatiles[16]. Information on the functional properties of an OR repertoire from an herbivorous species is needed not only to validate this hypothesis at the molecular level, but also to better understand the evolution of olfactory capacities and food preferences in insects.

Here we present a systematic functional analysis of a large array of ORs from an herbivorous moth, the cotton leafworm Spodoptera littoralis, and address the evolution of OR function in the order Lepidoptera. Spodoptera littoralis is a highly polyphagous noctuid whose larvae are pests of more than 80 crops in Africa, the Middle East and the Mediterranean basin[17]. It has been established as a model in the fields of chemical ecology and neurobiology of olfaction, and its antennal transcriptome was one of the first to be sequenced in insects[18]. We expressed 35 candidate S. littoralis ORs (SlitORs) in Drosophila OSNs using the empty neuron system[19] and identified SlitORs tuned to a variety of odorant molecules previously demonstrated to be physiologically or behaviourally active in this species[20–25]. We also provide the first evidence of a functional clustering of receptors within the phylogeny of lepidopteran ORs, with receptors from most basal lineages responding to aromatic compounds, whereas response spectra in more recently emerged clades are dominated by terpenes or aliphatics.

## Results

### The Drosophila empty neuron works well for non-dipteran ORs.

To provide a better view on the evolution of lepidopteran ORs, we investigated the function of a large array of ORs in a single species. Previous analyses of the S. littoralis adult transcriptome led to the identification of 47 candidate SlitORs, among which 35 had a full-length sequence[18,26,27]. As a first step, we built a phylogeny of lepidopteran ORs using S. littoralis receptors, together with OR repertoires identified in seven other lepidopteran species from seven different families (Fig. 1). In this tree, ORs fell into 21 highly supported clades (lettered A–U) representing the different lepidopteran OR lineages that evolved from ancestral genes. The 35 SlitORs belonged to 17 of the 21 lineages, thus ensuring a good cross-section of the OR diversity (Fig. 1). We generated transgenic fly lines, each expressing one of the SlitORs within ab3A OSNs instead of the endogenous Drosophila receptor[19]. Using RT–PCR, we confirmed correct expression of the SlitOR transgene in the antennae of 30 of the 35 lines (Supplementary Table 1); single-sensillum recordings were then performed on ab3 sensilla of theses 30 lines. For six lines, ab3A neurons displayed an abnormal spontaneous firing rate, with bursts of action potentials (Supplementary Fig. 1a). This phenotype was reminiscent of that observed in mutant ab3A neurons expressing no receptor[19] and indicated that these SlitOR transgenes, while expressed, were likely non-functional. The remaining 24 SlitORs appeared to be functional when expressed in ab3A OSNs, since they conferred regular spontaneous background neuronal activity (Supplementary Fig. 1b,c). Such a success rate (68% of functional ORs) is similar to what has been obtained with A. gambiae ORs using the same expression system[13].

Next, we systematically investigated the odorant detection spectrum of these 24 SlitORs using a panel of 51 volatiles presented at a high dose. These chemically diverse odorants were chosen based on their effect on the physiology or the behaviour of S. littoralis and other moths, and include host plant and herbivore-induced volatiles, oviposition cues, larval frass volatiles and pheromone components (Supplementary Table 2). Almost 10% of the tested OR-odorant combinations produced responses that differed statistically from the response to the solvent (Kruskal–Wallis ANOVA followed by Dunn's post hoc test, $P < 0.001$), distributed across 20 of the 24 tested SlitORs (Fig. 2a and Supplementary Fig. 2). Interestingly, we observed only excitatory responses and not any inhibitory response, which is consistent with the current knowledge of functional properties of moth OSNs[21,28]. Four SlitORs (5, 9, 21 and 30) did not display any significant response to stimuli, which may be linked to the low spontaneous firing rate observed (Supplementary Fig. 1b,e) or simply to the absence of relevant ligands in the test panel. Three other receptors (SlitOR10, 22 and 26) gave, at most, significant yet very low responses ($< 30$ spikes s$^{-1}$) and they were consequently not considered to be deorphanized in any biologically meaningful sense and not included in the analyses that follow. A summary of the 'success rate' of SlitOR expression is illustrated in Supplementary Fig. 1d.

To determine whether the empty neuron is a faithful expression system for SlitORs, we compared the SlitOR response profiles obtained with those of OSNs previously characterized in S. littoralis females[21]. At least six OR profiles clearly matched OSN profiles (Supplementary Fig. 3), despite the fact that the sets of SlitORs and S. littoralis OSNs compared here were partial. Furthermore, the response spectra of two sex pheromone receptors characterized here (SlitOR6 and SlitOR13) were previously shown to correlate with S. littoralis male OSNs[29]. Other deorphanized SlitORs may correspond to yet uncharacterized OSNs or, alternatively, OSN response profiles

may lack ligands included in the present panel. In any case, these correlations, together with the rate of successful expression, demonstrate that expression in *Drosophila* OSNs can be effectively used to carry out large-scale studies of ORs from outside the dipteran order.

**SlitORs exhibit a diversity of detection spectra.** The 17 deorphanized SlitORs presented a large diversity of response spectra at high-stimulus doses, regarding both the nature and the number of ligands identified. Thirty-two odorants from the three chemical classes tested (aliphatics, aromatics and terpenes) elicited significant responses from at least one SlitOR, with three ORs activated per ligand on average (Fig. 2a and Supplementary

Fig. 2). Aliphatic and aromatic compounds were detected by 16 and 13 SlitORs, respectively, whereas terpenes were detected by five ORs (SlitOR3, SlitOR4, SlitOR7, SlitOR29 and SlitOR35). On the other hand, most SlitORs were activated by several ligands from different chemical classes (six ligands per OR on average), consistent with a combinatorial model of odour coding[30]. There was a considerable degree of redundancy among OR detection spectra at this higher screening dose, with several ORs activated by a partially overlapping spectrum of aromatics and short-chain aliphatic alcohols (the so-called green leaf volatiles).

To provide a better view of the specificity of SlitORs, we built tuning curves (Fig. 2b) that represent the distribution of responses obtained for each SlitOR to the odorant panel and we calculated the sparseness of the distribution as a measure of

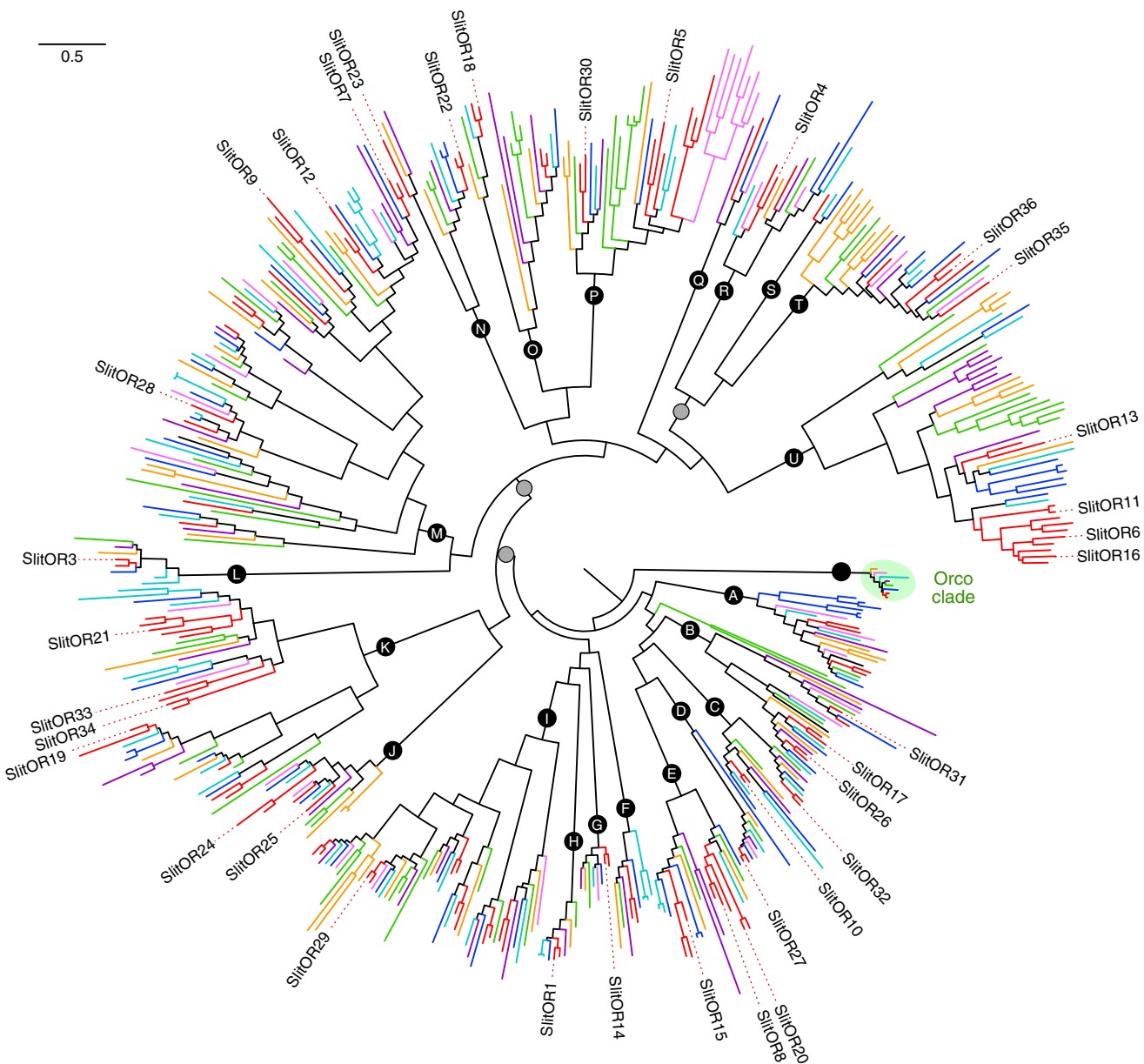

**Figure 1 | Maximum-likelihood phylogeny of lepidopteran ORs.** The phylogenetic positions of the 35 SlitORs whose function has been investigated are indicated. The tree was built from amino-acid sequences of OR repertoires of *S. littoralis*, *Helicoverpa armigera* (family Noctuidae, branches coloured in red), *Bombyx mori* (Bombycidae, dark blue), *Manduca sexta* (Sphingidae, light blue), *Dendrolimus houi* (Lasiocampidae, pink), *Heliconius melpomene* (Nymphalidae, orange), *Ostrinia furnacalis* (Crambidae, purple) and *Epiphyas postvittana* (Tortricidae, green). The Orco clade was used as an outgroup. Circles represent nodes highly supported by the likelihood-ratio test (black: aLRT >0.95; grey: aLRT >0.9). The letters in the circles (A–U) appoint the 21 OR clades. The scale bar represents 0.5 expected amino-acid substitutions per site.

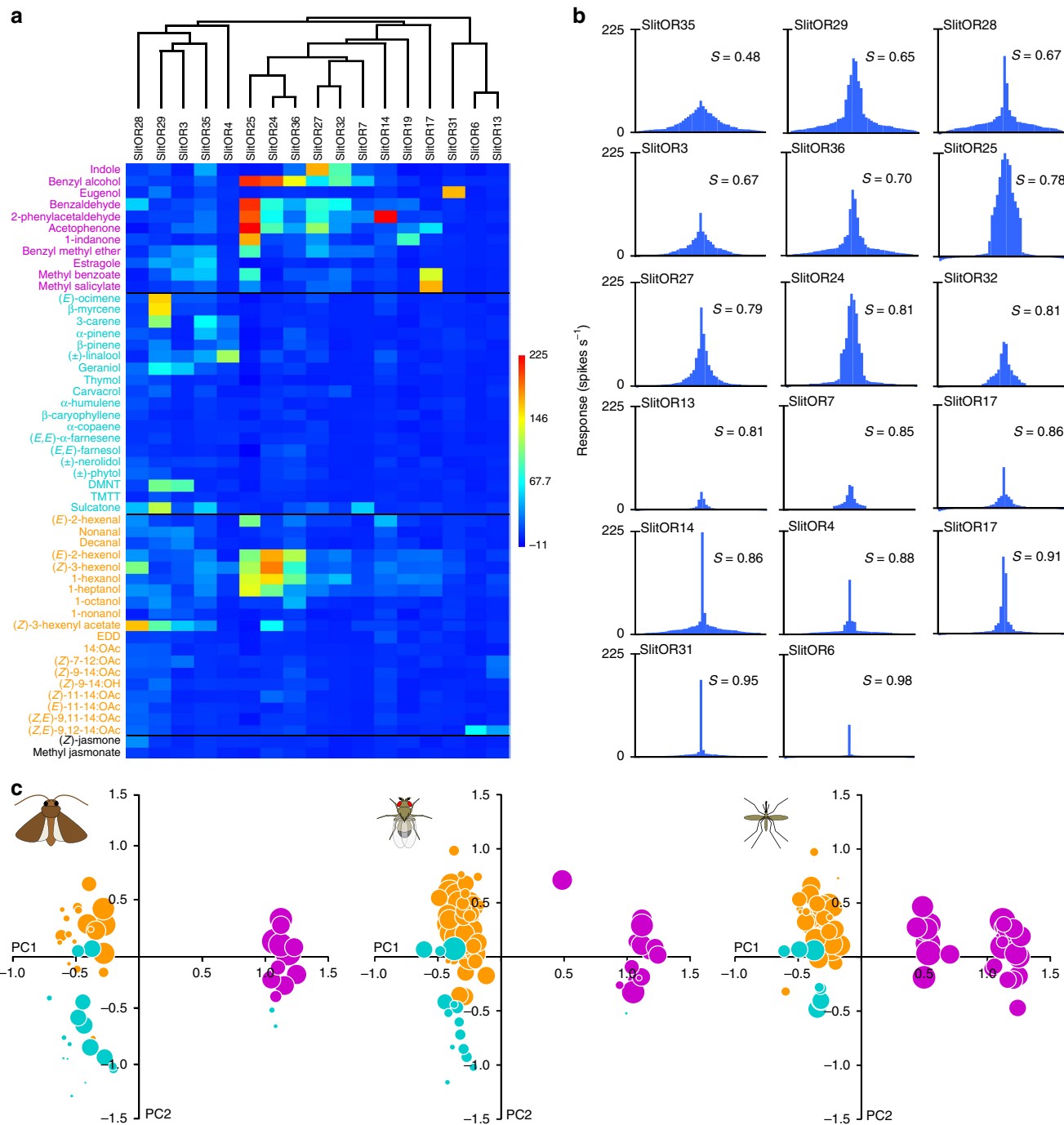

**Figure 2 | SlitOR response spectra at high-stimulus doses.** (**a**) Heat map summarizing the mean responses ($n = 5$ or 10) of 17 SlitORs to the panel of 51 odorants at high dosage (10 μg per pipette for pheromone compounds, 100 μg for the other compounds). Responses are colour-coded according to the scale on the right (firing rate in spikes s$^{-1}$). SlitORs are classified based on a cluster analysis of response spectra (see dendrogram at the top) and odorants are classified depending on their chemical class (magenta, aromatics; cyan, terpenes; orange, aliphatics; black, unclassified). (**b**) Tuning curves of SlitORs, showing the distribution of mean responses ($n = 5$ or 10) to the panel of 51 odorants at high dose. The tuning breadth of each receptor is represented by the sparseness value of the distribution ($S$)[31]. A low $S$ value indicates a broad tuning and a value of 1 indicates a narrow tuning of the receptor. (**c**) Bubble plots representing the distribution of OR responses in *S. littoralis*, *D. melanogaster* and *A. gambiae* among a virtual odour space based on physicochemical properties of the molecules. Odorants are distributed along the first two components from a PCA of the 32 normalized molecular descriptors described in ref. 62 and the size of each bubble is scaled to the capacity of the considered OR repertoire to detect the odorant. Colours represent the different chemical classes, as in **a**.

the OR tuning breadth[31]. We observed a continuum from very broad to very specific tuning. The most broadly tuned receptor was SlitOR35 (significant responses to 16 odorants), with a sparseness value of 0.48. At the other extreme, we observed that a large number of SlitORs exhibited a very narrow response spectrum, especially ORs that responded to ecologically relevant signals. For instance, SlitOR13 and SlitOR6, both detecting sex pheromone components, were narrowly tuned to their ligands,

thereby confirming previous results on pheromone detection in *S. littoralis*[29,32]. The other receptors exhibiting the narrowest tunings (that is, SlitOR4, 14, 17 and 31) were activated by plant volatiles commonly emitted by flowers (linalool, 2-phenylacetaldehyde, methyl salicylate and eugenol, respectively).

To compare the olfactory detection capacities of the present repertoire of SlitORs with those of *D. melanogaster* and *A. gambiae* ORs[10,13], we built bubble plots representing the global response of each OR repertoire across the same virtual odour space, a representation previously used by Carey *et al.*[13] (Fig. 2c). In each graph, dots represent the odorants eliciting a response from the OR repertoire: the position of a dot in the two dimensions is dependent on the physicochemical properties of the molecule, while the area of a dot reflects the capacity of the repertoire to detect the odorant. Whereas this presentation is, at this point, limited because of different ligand panels tested (for example, some aromatics have been tested only on mosquito, and fewer aliphatics have been tested on SlitORs), we felt this analysis might nevertheless reveal interesting features. Notably, although *S. littoralis*, *A. gambiae* and *D. melanogaster* OR repertoires seemed similarly effective at detecting the aromatic compounds (right part of the plots), monoterpenes (lower part of the plots) appeared to be better detected by the SlitOR repertoire than the *D. melanogaster* OR repertoire.

**SlitORs are narrowly tuned at low concentrations.** As previous observations related only to high-dose stimuli, we next investigated how the detection capacities of SlitORs would change according to the quantities of odorants. We conducted a systematic dose–response analysis for the 14 SlitORs that displayed strong responses ($>50$ spikes s$^{-1}$) at high dose of the odorants. To compensate for differences in volatility, we calculated and used, for each stimulus concentration, an estimate of the airborne quantity of molecules flowing out of the stimulus cartridge[33]. We first observed that some SlitORs exhibited particularly high sensitivity (Fig. 3). For example, OSNs expressing SlitOR14, SlitOR24, SlitOR25, SlitOR27 or SlitOR29 were still activated when challenged with less than 1 pmol of molecule flux (Kruskal–Wallis ANOVA followed by Dunn's *post hoc* test, $P<0.01$). Second, we observed for most SlitORs an apparent increase in response specificity when lowering the dose. One of the ligands generally appeared more potent than the others, that is, it was active at a considerably lower dose (Fig. 3).

To compare SlitOR responses to the exact same amounts for all odorants, we calculated the theoretical OSN firing rate for each SlitOR in response to a range of odorant flux from 10,000 down to 0.01 pmol, using linear regression. To assess the variation of SlitOR tuning breadth along this range, we plotted the sparseness of the OR response spectra for each odorant dose (Fig. 4a). As expected, the specificity of the SlitORs increased as the estimated airborne odorant dose decreased. The heat map of the theoretical responses to 100 pmol of odorants (which corresponds to the middle of the range of doses tested for most OR-ligand couples, as seen in Fig. 3) illustrates that 8 of the 14 receptors remained activated by only one ligand at this dose, and two other ORs were activated by only two ligands each (Fig. 4b). Notably, one-to-one relationships were found for four receptors (SlitOR31, SlitOR27, SlitOR17 and SlitOR4) and their four respective ligands. The 25 odorants still active on SlitORs at 100 pmol notably include 8 of the 12 more widespread flower volatiles[34], such as benzyl alcohol, methyl salicylate, (*E*)-ocimene or linalool and herbivore-induced volatiles known to elicit specific behaviour in *S. littoralis*, such as indole, DMNT or (*Z*)-3-hexenyl acetate (Supplementary Table 2).

Using the same dose–response data, we also plotted the number of ORs activated by each odorant across the range of doses tested. Notably, this number increased gradually with the dose for several odorants, irrespective of their chemical class (Supplementary Fig. 4). Taken together, our observations are in accordance with the current view that, at the peripheral level, combinatorial coding is likely to play a role in coding the variation of odorant quantity.

**A scenario of OR functional evolution in Lepidoptera.** Thanks to the present results on *S. littoralis* ORs, receptors have now been deorphanized in 13 different clades of the lepidopteran OR phylogeny, including previous results obtained from *Bombyx mori* and other moths[35–43]. To investigate the evolution of OR function in this insect order, we therefore placed these functional data in a phylogenetic framework, although it must be noted that these ORs have been deorphanized using different expression systems and different odorant panels. In the lepidopteran OR phylogeny (Fig. 5), clades A–K corresponded to the most basal lineages and they generally exhibited low mean genetic distances (Supplementary Fig. 5), as illustrated by short branch lengths in the tree. Deorphanized receptors from these clades were best activated by aromatics, with the exception of clades I and K that contained receptors to terpenes but also showed a much higher evolutionary rate that the other basal clades. The more recently emerged clades L–U formed a monophyletic group (Fig. 5) and deorphanized receptors from these clades had a terpene or an aliphatic as their best ligand. SlitOR36 represents the single exception, but this receptor exhibited high response thresholds towards all active ligands, suggesting that its key ligand(s) remains to be identified. In addition, receptors to aliphatics belonged to the OR lineages with the highest mean genetic distances (Supplementary Fig. 5). This suggests that receptors to aromatics emerged first and have been more conserved during the evolution of Lepidoptera, whereas receptors to terpenes and aliphatics emerged more recently and evolved faster (especially aliphatic receptors, which include pheromone receptors).

**Discussion**

We have investigated the molecular basis of olfactory reception in an herbivorous insect, the crop pest moth *S. littoralis*, by the systematic functional characterization of a repertoire of ORs representative of the diversity of lepidopteran ORs. We found that SlitORs presented a large diversity of response spectra, with more than 30 ligands identified from the three chemical classes tested, and a diversity of tuning breadths, from narrowly to more broadly tuned receptors. SlitORs appeared particularly effective at distinguishing short-chain aliphatic alcohols (also called green leaf volatiles), aromatics and terpenes, with the SlitOR repertoire being even more potent than the *Drosophila* repertoire at detecting monoterpenes. These properties correlate with the ecological needs of herbivorous and nectar-feeding insects[16]— such as moths—since aromatics and monoterpenes are the major constituents of plant odours emitted by both flowers[34] and leaves.

Interestingly, most SlitORs bound only one or two closely related odorants when tested at minute amounts, quantities frequently encountered by insects in nature while orienting over long distances. In Diptera, narrowly tuned receptors are often strikingly associated with odours of high biological salience. In *D. melanogaster*, two receptors are specific for cVA, a volatile sex pheromone[44,45], and several other ORs appear to be the unique detectors for non-pheromonal odours and are necessary and sufficient for the completion of vital behaviours. These include avoidance of toxic microbes by means of a mould odour[46] or oviposition decisions by means of fruit volatiles[47,48] and yeast metabolites[49]. In mosquitoes, highly selective receptors for human emanations used as kairomones for host recognition

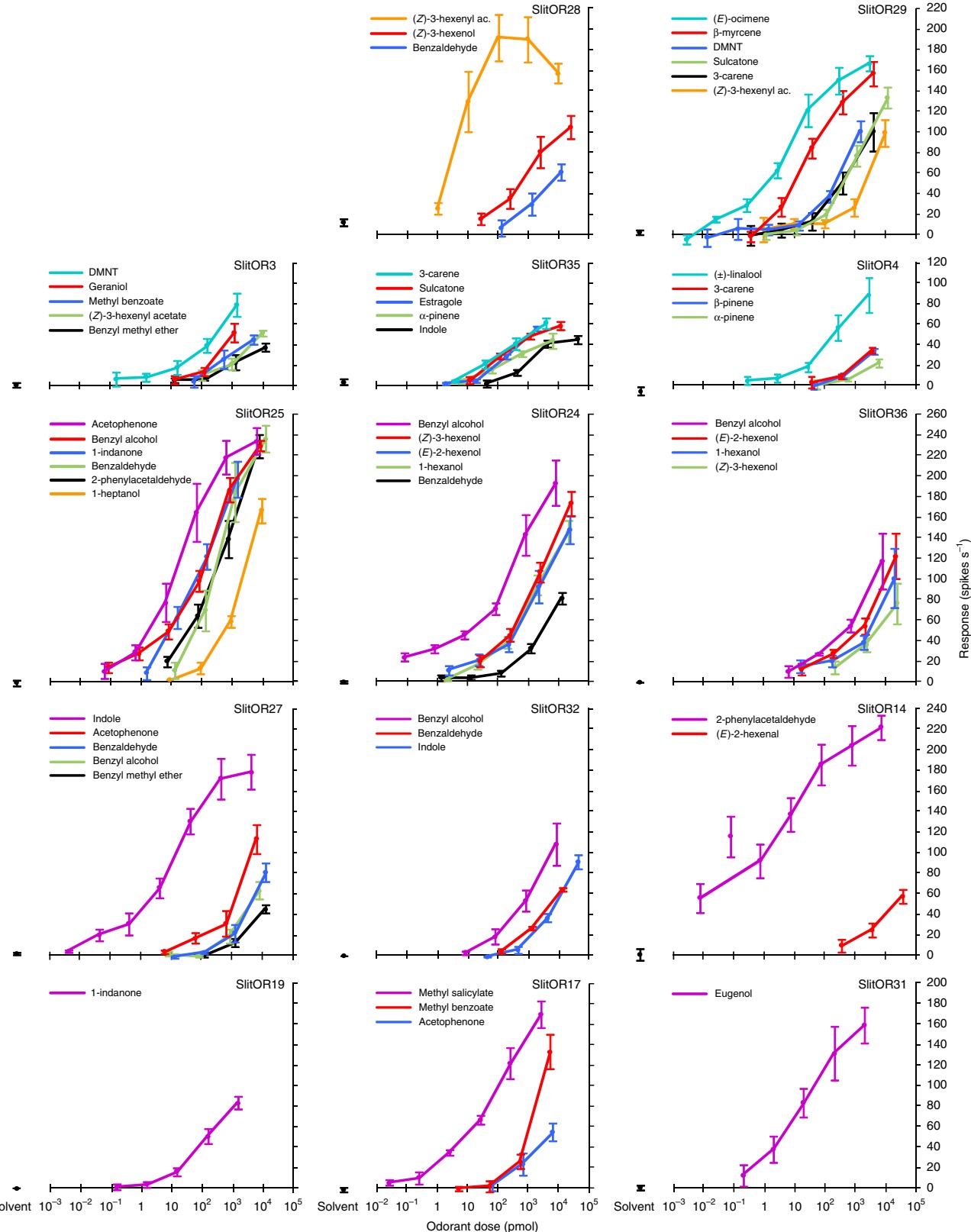

**Figure 3 | Dose–response analysis of SlitORs.** Mean response of 14 SlitORs to a range of odorant doses. Odorant doses correspond to the amount of molecules (in pmol) flowing out of the stimulus cartridge, to compensate the differences of volatility and affinity to paraffin oil solvent between chemicals[33]. Error bars indicate s.e.m. ($n = 5$). Responses to solvent are indicated in black. Only the six best ligands are represented for receptors on which more odorants have been tested.

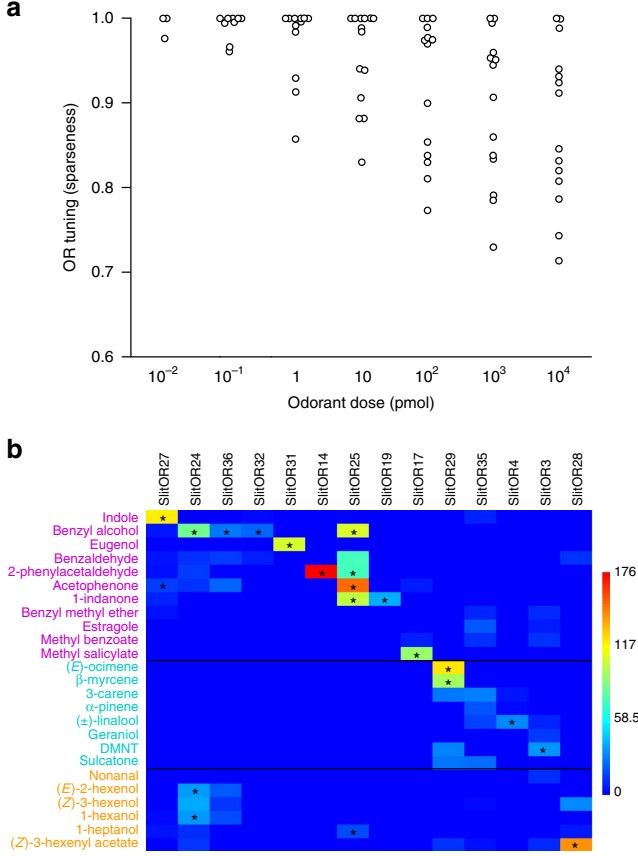

**Figure 4 | Dose-dependent SlitOR tuning breadths and response distributions across the repertoire.** (**a**) Variation of SlitOR tuning breadth across a range of odorant airborne quantities. Each point indicates the tuning breadth of one SlitOR at a given dose, represented by the sparseness value of the response distribution[31]. A low sparseness value indicates a broad tuning and a value of 1 indicates a narrow tuning of the receptor. (**b**) Heat map of the responses of SlitORs to 100 pmol of odorants. Response rates were predicted based on linear regression equations, and are colour-coded according to the scale on the right (spikes s$^{-1}$). *: OR-ligand couple for which responses to less than 100 pmol are statistically different from 0 (Kruskal–Wallis ANOVA followed by a Dunn's *post hoc* test, $P < 0.001$, $n = 5$).

and blood feeding or for oviposition cues have also been found[13,14,50–52]. In moths, such specific pathways have been very well described for sex pheromone perception, but it remains unclear whether they also exist for salient plant odorants. Electrophysiological studies performed on *S. littoralis* and a range of herbivorous insects demonstrate that they are all equipped with numerous sensory neurons narrowly tuned to common plant volatiles[16,21,23,53–55]. Although some specialist species detect compounds specific to a narrow range of plant taxa, a majority of herbivorous insects discriminate host from non-host plants through specific combinations of more ubiquitous odorants. This discrimination capacity could be of particular relevance for polyphagous feeders such as *S. littoralis*. Here at low doses, we found 10 SlitORs tuned to a limited number of odorants in our panel, all of which are widely occurring odorants emitted by numerous flowering plants[34]. These ubiquitous odorants are components of plant headspaces or synthetic blends already proven to activate behaviours of *S. littoralis*, but they have little effect when presented individually[22,24,25,56]. Rather, behavioural activity towards salient odorant blends would arise from concomitant activation of separate input channels (that is,

SlitORs). Furthermore, the increasing number of ORs activated when raising the odorant dose, as observed here with SlitORs, could constitute the molecular basis for a combinatorial coding of the odorant quantity. Previous work on *D. melanogaster* larval behaviour provided evidence that all the receptors detecting a single, unique odorant at various concentrations are indeed necessary for the achievement of the behaviour towards the compound across the range of concentrations[11,57]. Thus, having several receptors to the same odorant but with different detection thresholds would allow for a precise coding of the odorant quantity over a large range of concentrations, which is critical for herbivorous insects to successfully discriminate specific plant odorant mixtures.

The phylogeny of lepidopteran ORs suggests that advanced lepidopteran species (Ditrysia) share the same basic groups of ORs, which evolved from a limited number of ancestral genes. This observation is consistent with the low number of lineage-specific expansions observed among lepidopteran ORs, and the fact that they evolved under strong purifying selection[58]. The radiation of Ditrysia appears to coincide with the radiation of angiosperms[59], suggesting that Ditrysia ORs could have evolved in tandem with early floral diversification. Strikingly, the combination of functional and phylogenetic analyses we performed on lepidopteran ORs shows a basic major division in the phylogeny, where receptors for aromatics are housed in basal clades of the tree and are the most conserved, whereas sex pheromone receptors and a large part of ORs tuned to terpenes and short-chain acetates generally belong to later lineages with a higher rate of evolution. Hence, these OR lineages are more likely to be involved in the adaptation of the chemosensory system of these animals to new ecological feeding niches.

Deorphanization of additional ORs from *S. littoralis* and other herbivorous insects, as well as functional assays carried out with larger panels of odorants, are expected to complement and strengthen the findings from the current study. Here we expressed SlitORs in the empty neuron system, which has been used previously for studying large OR repertoires in only the dipterans *D. melanogaster*[8–12] and *A. gambiae*[13]. Although we show that the empty neuron is suitable for deorphanizing ORs from outside Diptera, some SlitORs were not functional in this *in vivo* expression system and may need important factors necessary for correct functioning that are lacking in ab3 sensilla. For instance, at1 neurons constitute a better expression system for a *Drosophila* terpene receptor as well as some moth pheromone receptors, possibly due to the presence of sensory neuron membrane proteins[32,47,60]. Alternatively, we cannot exclude that these SlitORs are in fact non-functional *in natura*. Another limitation of large-scale studies of ORs is the number of candidate ligands included in functional assays. In this regard, the development of more high-throughput *in vitro* assays should allow a significant increase in the number of odorant molecules tested and a reduction in the portion of remaining orphan receptors. Moreover, standardized assays conducted with similar sets of odorants will encourage more in-depth comparisons between OR repertoires of different insects.

In conclusion, we have performed a systematic functional characterization of a high number of ORs from a lepidopteran species, and have also linked their function to their evolutionary history. As such, our study provides not only a reference for further investigations of the molecular bases of olfaction in Lepidoptera, but it also lays the foundations to understand how a polyphagous herbivore uses a receptor array in the complex task of selecting a host plant among many potential hosts. Our work also opens up new routes for crop pest control, since narrowly tuned receptors involved in vital behaviours have been identified. Such receptors appear to be particularly good targets for the

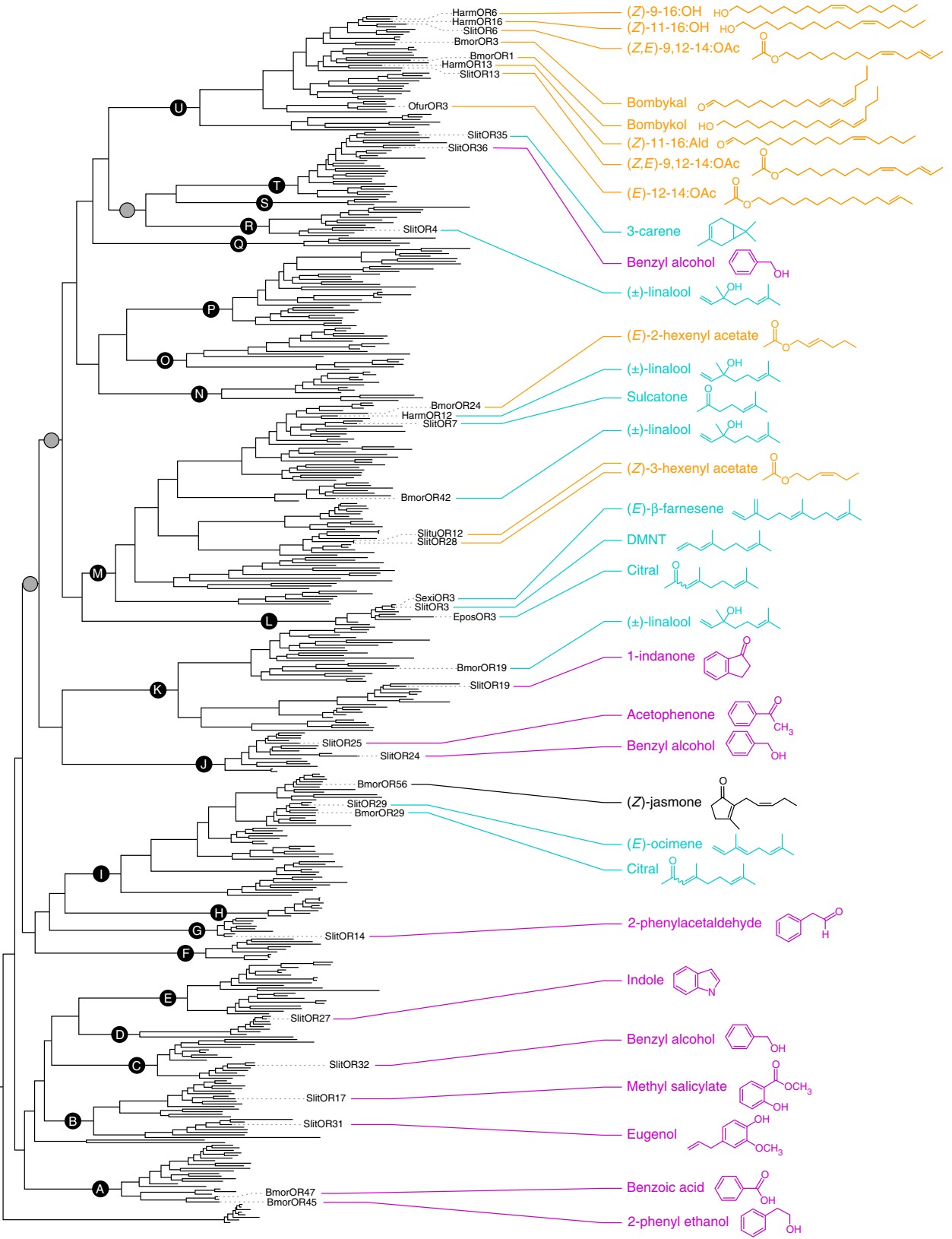

**Figure 5 | Functional evolution of lepidopteran ORs.** Maximum-likelihood phylogeny of lepidopteran ORs, highlighting the ligands identified. The phylogeny is the same as in Fig. 1. Circles represent nodes highly supported by the likelihood-ratio test (black: aLRT > 0.95; grey: aLRT > 0.9) and the letters in the circles appoint the 21 OR clades. For each functionally characterized receptor, only the ligand eliciting the highest response in the corresponding functional assay is indicated, based on results from the present study and from previous works on ORs expressed in *Xenopus* oocytes (*S. litura*[40], *S. exigua*[38], *H. armigera*[43,70], *B. mori*[35,37,41] and *O. furnacalis*[42]) and Sf9 cell lines (*B. mori*[38] and *E. postvittana*[36]). Colours of each ligand represent the different chemical classes, as in Fig. 2.

screening of activators and/or inhibitors that would interfere with host plant finding.

## Methods

**Generation of UAS-SlitOr constructs.** For the generation of pUAST and pUAST.attB constructs, cDNA fragments were first cloned into pCRII-TOPO (Invitrogen, Carlsbad, CA, USA), sequenced, and subcloned into destination vectors using the appropriate restriction enzymes. For the generation of pUASg-HA.attB vectors, cDNA fragments were cloned into the pCR8/GW/TOPO Gateway Entry vector (Invitrogen), sequenced, and subcloned into pUASg-HA.attB via Clonase II-mediated enzymatic transfer (Invitrogen). Before injection into Drosophila embryos, plasmid constructs were purified from liquid cultures of One Shot TOP10 E. coli using the EndoFree Plasmid Maxi kit (Qiagen, Venlo, Netherlands).

**Heterologous expression of SlitORs in Drosophila.** For P-element transgenesis, pUAST-SlitOR plasmids were injected into $w^{1118}$ fly embryos, and fly lines harbouring a transgene insertion into the third chromosome were used for further crossings. For phiC31-targeted transgenesis, pUAST.attB-SlitOR or pUASg-HA.attB-SlitOR plasmids were injected into embryos with the genotype $y^1$ M{vas-int.Dm}ZH-2A w*; M{3xP3-RFP.attP}ZH-86Fb, leading to an insertion of the UAS-SlitOR constructs into the genomic locus 86Fb of the third chromosome (see Supplementary Table 1 for the procedure followed for each OR). Flies were reared on standard cornmeal-yeast-agar medium and kept in a climate- and light-controlled environment (25 °C, 12 h light: 12 h dark cycle). The presence of the UAS-SlitOR transgenes was verified by PCR on genomic DNA extracted from two flies, and the expression of the SlitORs in antennae was verified by RT–PCR on total RNA extracted from ≥100 pairs of antennae.

**Single-sensillum recordings.** For all experiments, a randomly chosen fly within a strain (female, 2- to 6-day-old) was restrained in a plastic pipette tip with only the head protruding from the narrow end. The pipette tip was fixed with dental wax on a microscope glass slide with the ventral side of the fly facing upward. Then, the antenna was gently placed on a piece of glass slide and maintained by placing a glass capillary between the second and third antennal segments, held in place by dental wax. Afterwards, the slides were placed under a light microscope (BX51WI, Olympus, Tokyo, Japan) equipped with a × 50 magnification objective (LMPLFLN 50X, Olympus) and a × 15 eyepieces. Desiccation of the flies was avoided with the help of a constant $1.5 \, l \, min^{-1}$ flux of charcoal-filtered and humidified air, delivered through a glass tube of a 7 mm diameter, which terminated ca. 1.5 cm from the antenna.

Stimulation cartridges were built by placing a 1 cm² filter paper in the large opening end of a Pasteur pipette and dropping 10 µl of an odorant solution onto the paper before closing the pipette with a 1 ml plastic pipette tip. Odorant stimulations were performed by inserting the tip of the Pasteur pipette into a hole in the glass tube and generating a 500 ms air pulse ($0.6 \, l \, min^{-1}$), which reached the permanent air flux ($1.5 \, l \, min^{-1}$) while going through the stimulation cartridge.

Action potentials were recorded from ab3 sensilla using electrolytically sharpened tungsten electrodes. The reference electrode was inserted into the eye of the fly by a manually controlled micromanipulator. The thinner recording electrode was inserted at the base of the sensillum of interest using a motor-controlled DC-3K micromanipulator (Märzhäuser, Wetzlar, Germany) equipped with a PM-10 piezo translator (Märzhäuser). The electrical signal was amplified using a UN-06 AC-DC amplifier (Syntech, Kirchzarten, Germany), digitized through an IDAC-4-USB (Syntech) then recorded and analysed using Autospike (Syntech), or amplified using an EX-1 amplifier (Dagan, Minneapolis, MN, USA), digitized through a Digidata 1440A acquisition board (Molecular Devices, Sunnyvale, CA, USA) then recorded and analysed using the pCLAMP 10 software (Molecular Devices). The net responses of ab3A neurons expressing a SlitOR were calculated by subtracting the spontaneous firing rate from the firing rate during the odorant stimulation. The time windows used to measure these two firing rates lasted for 500 ms and were respectively placed 500 ms before and 100 ms after the onset of stimulation. This 100 ms time lag was defined to take into account the time for the odorants to reach the antenna. For pheromone compounds (which have a lower volatility and reached the antenna later), the response counting window was shifted to begin 400 ms after the onset of the stimulation.

As is it not possible to distinguish ab3 sensilla only by localization and morphology, 100 ng of 2-heptanone were used as a diagnostic stimulus, since this odorant is one of the most potent ligands for DmelOR85b, which is expressed in the ab3B neuron[10]. The absence of DmelOR22a in ab3A neurons expressing a SlitOR was verified using a stimulus cartridge containing 100 ng of ethyl hexanoate, a strong ligand for DmelOR22a[10].

**Odorant stimuli.** Response spectra of ab3A neurons expressing SlitORs were tested against a panel of 51 odorants (Supplementary Table 2). Pheromone compounds were used at a $1 \, \mu g \, \mu l^{-1}$ dilution in hexane (10 µg deposited on the filter paper). Other compounds were used at a $10 \, \mu g \, \mu l^{-1}$ dilution in paraffin oil (100 µg on the filter paper), apart from indole, which was also diluted in hexane.

The stimulation cartridges were used at most two times on each fly and a maximum of five times in total. Pipettes with filter papers containing 10 µl of solvent were used as controls. For each receptor, the entire odorant panel was tested five times on five different flies. Odorants were considered as active if the response they elicited was statistically different from the response elicited by the solvent alone (Kruskal–Wallis ANOVA followed by a Dunn's post hoc test, $P < 0.001$). Then, for each receptor, odorants considered as active were tested five more times.

**Data analysis.** Statistical analyses were performed with Prism (GraphPad Software, La Jolla, CA, USA). Heat maps, principal component analyses (PCA) and measures of Euclidean distances were performed using the PAST 3 software[61]. Bubble plots were obtained through a PCA (using the variance-covariance matrix) of an optimized set of 32 molecular descriptors[62] for 154 odorants tested on S. littoralis, D. melanogaster or A. gambiae ORs[10,13]. Classes that have not been tested on SlitORs (that is, sulfurs, lactones, acids and amines) were excluded from this analysis. Molecular descriptors were obtained with the Dragon 6 software (Talete, Milano, Italy) and were normalized[13] to their respective maximum value. For each odorant tested on a given OR repertoire, the area of the dot was scaled to the Euclidean distance (in a space where the response of each OR in spikes $s^{-1}$ is a dimension) between the response to the odorant and a null response across the repertoire. To take into account differences in the range of response amplitudes between OR repertoires from different species, Euclidean distances were normalized to the maximum distance measured for each repertoire.

The sparseness of the SlitOR response spectra was calculated using the formula of Rolls and Tovee[31]: $S = (\sum_{i=1,n} r_i/n)^2 / \sum_{i=1,n} (r_i^2/n)$ with $r_i$ being the firing rate to the stimulus $i$ in the set of $n$ stimuli. As this formula cannot compute negative responses, they were set to 0. We used standard subroutines of IBM SPSS v21 for hierarchical clustering of SlitOR response profiles by CLUSTER with default settings (Squared Euclidean distance, Average linkage between groups), using solvent response corrected responses (spiking frequency of SlitOR-expressing ab3A neurons) at $n > 5$ per OR type and stimuli for an initial check that recorded cells clustered with their designated type. For display of SlitOR responses per stimulus, we used the mean value per OR type. Before clustering of both individual cell data and their means, cases were standardized to Z scores (to mean 0 and an s.d. of 1 by subcommand PROXIMITIES), to compensate for differences in absolute activities recorded from each OSN.

**Phylogeny.** To take into account the taxonomic diversity of Lepidoptera, the OR phylogeny was built from OR sequences from seven different lepidopteran families. For each family, the species in which the OR repertoire was best annotated was chosen. The data set contained 469 OR amino-acid sequences from S. littoralis[27], Helicoverpa armigera[63] (Noctuidae), Bombyx mori[35,37,41] (Bombycidae), Manduca sexta[64] (Sphingidae), Dendrolimus houi[65] (Lasiocampidae), Ostrinia furnacalis[66] (Crambidae), Heliconius melpomene[67] (Nymphalidae) and Epiphyas postvittana[68] (Tortricidae), plus two sequences of ORs deorphanized in Spodoptera litura[40] and Spodoptera exigua[38] (Noctuidae). Sequences were aligned with MAFFT v.7 (http://mafft.cbrc.jp/alignment/server/). The maximum-likelihood tree was built with PhyML 3.0 (http://www.atgc-montpellier.fr/phyml/), using the JTT + G + F substitution model as determined by ProtTest 2.4 (http://darwin.uvigo.es/software/prottest2_server.html). Node support was estimated using a hierarchical likelihood-ratio test[69]. Phylogenetic distances were extracted from the maximum-likelihood tree using SeaView v.4 (http://pbil.univ-lyon1.fr/software/seaview).

**Data availability.** All relevant data are available on request to the corresponding authors.

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

## Acknowledgements

The authors thank John Carlson (Yale University) for providing the *Δhalo;Or22a-GAL4* fly line, for sharing raw SSR data on *D. melanogaster* and *A. gambiae* ORs and for fruitful discussions on the study; Johannes Bischof (University of Zurich) for providing the *pUAST.attB* and *pUASg-HA.attB* vectors; Philippe Lucas, Fabien Tissier and Philippe Touton (INRA Versailles) for the installation of the electrophysiology apparatus; Sébastien Fiorucci and Jérôme Golebiowski (University of Nice Sophia Antipolis) for providing molecular descriptors; Göran Birgersson (SLU Alnarp) for his help with the classification of chemicals, and the members of the Max Planck Institute for Chemical Ecology, notably Ewald Grosse-Wilde, Sonja Bisch-Knaden and Hany Dweck, for insightful comments and fruitful discussions. This work was funded by INRA, UPMC, the French National Research Agency (ANR-09-BLAN-0239-01, ANR-16-CE21-0002-01, ANR-16-CE02-0003-01), the Swedish Science Council (#621-2009-5160 and the Linnaeus initiative grant 'Insect Chemical Ecology, Ethology and Evolution' IC-E³ #217-2006-1750: The Swedish Research Council, FORMAS and SLU), the Trygger Foundation (CTS 10: 166), the Olle Engkvist foundation (2013.4.1-766), the Max Planck Society and the Higher Education Commission (HEC), Pakistan. The funders had no role in study design, data collection and analysis, decision to publish or preparation of the manuscript.

## Author contributions

A.d.F., W.B.W., N.M., F.S., P.A., B.S.H., M.C.L. and E.J.-J. designed research; A.d.F., W.B.W., N.M., C.S., M.B., T.C., A.M., M.-C.F. and C.M. performed the experiments; F.S., T.C., P.A. and B.S.H. contributed new reagents/analytic tools; A.d.F., W.B.W., N.M., M.B., F.S., T.C., M.C.L. and E.J.-J. analysed the data; A.d.F., W.B.W., N.M., F.S., M.C.L. and E.J.-J. wrote the paper with contributions from all authors.

## Additional information

**Competing interests:** The authors declare no competing financial interests.

