## [Peer Review File · Nature Communications]

Reviewers' comments:

Reviewer #1 (Remarks to the Author):

Overall, this is a well done study. A functional characterization of a large number of odor receptors of *Spodoptera littoralis* has never been completed before. This study will encourage further investigations into how an herbivorous insect uses its complement of odor receptors to select host plants for feeding and egg laying.

Below, I have listed some major and minor concerns with this ms. These include some questions/suggestions that the authors might consider to further improve the ms.

Major concerns:

1. Line 153: *While S. littoralis, A. gambiae and D. melanogaster OR repertoires were similarly efficient at detecting the aromatic compounds (right part of the plots), monoterpenes appeared to be better detected by the SlitOR repertoire than the D. melanogaster OR repertoire (lower part of the plots).*

I don't think there is enough information to make this conclusion. The way the bubble plots have been depicted could be misleading, especially given that different ligand panels were tested. How do you normalize the olfactory responses between different insect species, especially when the data for each was obtained in different laboratories and when the experiments were conducted by different individuals? Perhaps a better way to address this would be for the authors to undertake a select number of OR comparisons between species in their own lab, by the same individual, and for a set panel of terpenes.

2. Fig 4a. Not enough information is provided either in the results section or figure legends or materials and methods to be able to understand this graph. Moreover, the idea that an odor receptor specificity increases as concentration of odorant drops has been shown before for fly and mosquito receptors. While I acknowledge that this piece of information is new for Slit odor receptors, there is nothing novel about this concept.

3. Line 182: *"Taken together, our observations suggest that, at the peripheral level, combinatorial coding may not be the rule for coding odorant identity at low doses but is likely to play a role in coding the quantity of some odorants"*

In a similar vein as point #2, the above observation maybe be a new finding for Slit Ors but is not really a significant finding with regards to olfactory information processing.

4. Line 240: *Here, at low doses, we found 10 SlitORs narrowly tuned to widely occurring odorants, emitted by numerous flowering plants*

To claim that some receptors are very narrowly tuned and respond to only one or two odorants based on a very small panel (relative to other studies that have included much larger odor panels to test fly or mosquito odor receptors) is a reach.

5. Line 189: *We therefore placed these functional data in a phylogenetic framework, in order to investigate the evolution of OR function in this insect order.*

While I think this is an interesting analysis and useful conclusion, I wish the authors are a little more explicit in describing the nature of functional data obtained from other studies. It must be noted that the functional data for odor receptors in other moth species were obtained using different methodologies. Although the authors take a more empirical approach for this analysis and while I believe the results are valid, I think caveats must be noted, either in the results section or in the Discussion section.

Minor concerns/suggestions:

1. First paragraph of discussion is usually reserved for the most important findings of the study. Placing the finding "Slit Ors are functional in the empty neuron" in this paragraph takes a bit away from the other important findings of the study, especially for a journal such as Nature Communications.
2. Discussion section would benefit from a sub-section on any specific limitations and caveats regarding the conclusions of the current study

Reviewer #2 (Remarks to the Author):

The manuscript by Fouchier et al represents a very worthwhile contribution to a better understanding of the molecular basis of olfaction. The authors have performed the most extensive functional analysis of a large array of OR candidates in a non-model organism so far. The authors show that the empty neuron technique is suitable for characterization of olfactory receptors outside Diptera. A basic conservation of function within the receptor repertoire of Lepidoptera, across the evolutionary radiation of different major clades is highlighted. With no doubt this study will serve as a reference for further studies in this field.

The study is very well designed, taking advantage of a multitude of relevant techniques/approaches. The results appear solid and are discussed in a balanced way, and the manuscript is well written with nice and informative figures.

My comments are all relatively minor:

Lines 47-48: "paves the way for the development of innovative olfactory-based control strategies" – this is not very specific and could just as well be deleted. Not clear (not even in the main text) what the significant contribution to future applications in pest control is. I think that the authors should stick to the molecular basis of olfaction and refrain from claiming applied implications.

Lines 50-52: First sentence could be deleted. Just idling! Second sentence is a good enough (actually better) start of the introduction.

Lines 93-106: I think that this paragraph could be improved to more clearly describe how

the 47 OR candidates from the transcriptome were "sampled", which were functionally characterized and "what about the rest". This is partly dealt with in the discussion (lines 212-216) but I think that there is room for improvement.

- 47 candidates were identified in the transcriptome
- These candidates were distributed among 17 of the 21 highly supported clades in the phylogeny. Any comment on the fact that there were no Slit representatives in the other four clades?
- Then 35 (out of 47) of the candidates belonging to the 17 lineages were characterized. How were these 35 selected? Why were the other 12 discarded at this point?
- 5 of the 35 did NOT express in the *Drosophila* antenna? Any comment on why? Did this bias the results in any way?
- 6 out of the 30 were considered "non-functional". I would like to see a somewhat more extensive discussion of the criteria used and possibly some references to other studies (to help somebody not really familiar with the empty neuron system to understand what this "non-functionality" means).

Apart from that it is pleasing to find that the success rate was similar to what has previously been obtained in *A. gambiae*. Well done!

- Then four out of the 24 did not display any significant response to the stimuli. Isn't the most likely explanation that relevant ligands for these receptors were not included in the panel? The same may hold for the three that showed very low responses.

Line 123: I do not follow. Delete "only"?

Lines 148-152: I am a bit confused. The size of a bubble represents the global response of OR repertoires of the different species? But what determines the position of a bubble in the PCA plot? Does each bubble represent a compound? Maybe I am just ignorant but then I may still not be the only one. Please provide a better explanation.

Line 160: It is surprising to me that no ORs responding to pheromone components were included. I would imagine that the responses were quite high, especially if corrected for the relatively low volatility of these compounds compared to many of the plant compounds.

Lines 187-188: How were the species chosen to be included in the phylogeny? (see lines 360-363. No explanation for the selection). How robust is the topology of the tree if it is constructed with different species included? Please comment.

Lines 300-301: "equipped with a 50x magnification objective" but what about the total magnification? Ocular? 50x magnification appears low to me for this kind of work.

Lines 320-322: What was the counting window shifted from 100 ms to 400 ms for the pheromone compounds? Does not make any sense to be. Requires explanation.

List of references: May need some attention. Several scientific names are not in italics.

Figure legend 1: Add info on what the letters in the circles represent.

Figure 2a: The authors write (Z,E)-9,11-14:OAc but for instance (Z)9-14:OAc. I think it should be (Z)-9-14:OAc to be consistent. This discrepancy holds also for other compounds (e.g. (E)2-hexenal) and for supplementary table 2 where I also think that there should be room to spell out the full names of the pheromone components together with the short forms.

Figure 3: Why are OR13 and OR6 not included? Doses for the pheromone compounds in Fig 2A are 10x lower than for the other compounds and based on this I consider the responses in Fig 2A significant (and should qualify the receptors to be compared with the others in Fig 3).

Supplementary figure 1: It is not clear to me why (a) but not (b) represents a non-functional receptor. Please explain the criteria. Also explain the difference between the cells (different spike amplitudes). It is the cell firing with a large spike amplitude that is relevant in this study?

Supplementary figure 3: Improve the layout (introducing some space between the "pairs" to make it more clear what should be compared.

Reviewer #3 (Remarks to the Author):

Review of de Foucher et al.: "Functional evolution of Lepidoptera olfactory receptors revealed by deorphanization of a moth repertoire"

For: Nature Communications

This paper by de Foucher et al. is an important contribution to the field of insect olfaction, and to animal olfaction in general. The results are novel and should be of interest to readers interested not only in animal olfaction, but also to those interested in neuroethology, evolution, and animal behavior. A crucial finding in this study is that the *Drosophila* "empty neuron" technique can be successfully used for non-dipteran investigations of OR functionality, providing a robust platform for the de-orphanization of non-dipteran ORs. The finding that a good many of the OR response profiles matched nearly perfectly the response profiles of known, native *Spodoptera littoralis* olfactory sensory neurons (OSNs) is significant in that it further validates the novel use of the empty neuron technique in this study: i.e., the lepidopteran ORs are shown to behave normally in this dipteran expression system. The paper provides a first-ever basis for comparing odorant receptor (OR) function across the Lepidoptera for so-called "general odorants" comprising the vast majority of molecules encountered by insects in various environments. Moreover, the results strongly suggest a link between the importance of certain odorants with regard to the life histories of moth and fly species according to the narrowness of OR tuning profiles in response to these

ligands. To their credit, the authors use dose-response profiles and investigate responses to very low emitted amounts of odorants, thereby demonstrating the true narrowness of ligand specificity of a large number of ORs. The authors gamely construct a phylogeny of lepidopteran OR-ligand couplings along with the chemical classes of these ligands to then relate these patterns to the ORs that might be more recently evolved.

I have some minor editorial corrections/suggestions throughout the manuscript, plus two more major comments:

1) I wonder if there was never an inhibitory response to any of the odorants? I assume there wasn't because the profiles in Figure 2b do not show any negative values and the method for subtracting out the background firing rate from the post-stimulus-delivery firing rate would have revealed a sub-background level of firing that would have been displayed in Figure 2b. If this is so, I feel it would be helpful for the authors to state this outright somewhere, i.e., that there was never a reduction in background firing rate to any of the odorants tested. The narrow-vs.broad-tuning OR profiles in the similar histograms of the Hallem and Carlson *Drosophila* work nearly all exhibited a good proportion of negative values (inhibition) in response to some of the >100 odorants they tested. Their results may or may not represent a significant difference between lepidopteran ORs and *Drosophila* ORs' tuning characteristics, but maybe it does?

2) On lines 183 and 184, I think the statement that "combinatorial coding may not be the rule for coding odorant identity at low doses..." is not entirely correct. Combinatorial coding would be used even if only one glomerulus is activated by one specifically tuned type of OR/OSN because the activity of the neuropil in that one glomerulus is nevertheless being compared against the total inactivity of the other glomeruli. If activity in other glomeruli begins to occur at higher doses, the relative intensities of glomerular activity would still be centered around the higher activity in that first glomerulus. This is just part of a continuum of combinatorial coding of odor quality at very low vs. higher odorant intensities.

Minor comments, most having to do with readability:

Line 96: might be better to say, "In this tree, ORs fell into groups comprising 21 highly..". (To me, this sentence at first did not appear to have a verb; it took careful re-reading to realize that "clustered" was in fact the verb).

Line 151: may be better to say: "Whereas this presentation is at this point limited..."

Lines 152-153: I would suggest: "...only in mosquito and fewer aliphatics have been tested on SlitORs), we felt this analysis might nevertheless reveal interesting features."

Line 153: use "Although" instead of "While"

Line 154: use "effective" instead of "efficient"

Line 178: replace "on" with "in eliciting"

Line 195: replace "grouped" with "fell"

Line 209: use "because" rather than "as"

Line 211: replace "considered" with "compared"

Line 212: Wouldn't "functionally characterized" be more precise than "functionally expressed"?

Line 214: replace "and" with "as well as"

Line 216: I'm not sure of the technical terminology here, but would suggest that "inactive" might be a better word for better readability than "refractory".

Line 220: better to say, "...appeared particularly effective at responding to..."

Line 222: replace "one" with "repertoire" (even though "repertoire" will be used twice in this sentence)

Line 224: change to, "...plant odours emitted by both flowers and leaves."

Line 229: replace "whereas" with "and"

Line 230: remove comma

Line 238: replace "While" with "Although"

Line 239: better to say, ".....herbivorous insects discriminate host from non-host..."

Lines 240 and 241: remove commas

Line 247: replace "brought" with "provided"

Line 250: add "but", i.e., "the same odorant but with different detection thresholds..."

Line 251: replace "detect" with "characterize"

Line 272: replace "appear as" with "appear to be"

Reviewer #1 (Remarks to the Author):

Overall, this is a well done study. A functional characterization of a large number of odor receptors of *Spodoptera littoralis* has never been completed before. This study will encourage further investigations into how an herbivorous insect uses its complement of odor receptors to select host plants for feeding and egg laying.

Below, I have listed some major and minor concerns with this ms. These include some questions/suggestions that the authors might consider to further improve the ms.

Major concerns;

1. Line 153: *While S. littoralis, A. gambiae and D. melanogaster OR repertoires were similarly efficient at detecting the aromatic compounds (right part of the plots), monoterpenes appeared to be better detected by the SlitOR repertoire than the D. melanogaster OR repertoire (lower part of the plots).*

I don't think there is enough information to make this conclusion. The way the bubble plots have been depicted could be misleading, especially given that different ligand panels were tested. How do you normalize the olfactory responses between different insect species, especially when the data for each was obtained in different laboratories and when the experiments were conducted by different individuals? Perhaps a better way to address this would be for the authors to undertake a select number of OR comparisons between species in their own lab, by the same individual, and for a set panel of terpenes.

Answer:

Following the reviewer suggestion, we edited the paragraph to better explain how bubble plots were generated and what they represent (see lines 166-170 in the new text version). Details on Euclidian distance calculation and normalization procedure have been added in the Method part for clarification (see lines 384-389: "For each odorant tested on a given OR repertoire, the area of the dot was scaled to the Euclidean distance (in a space where the response of each OR in spikes.s⁻¹ is a dimension) between the response to the odorant and a null response across the repertoire. To take into account differences in the range of response amplitudes between OR repertoires from different species, Euclidean distances were normalized to the maximum distance measured for each repertoire". Such normalization allowed in part to overcome the biases that may arise from the fact that the studies have been conducted in different labs by different persons and on different species, but we agree our data still have to be taken with some caution and this has been added to lines 170-173. Moreover, as suggested later by the reviewer (see below), we included a paragraph on limitations and caveats in the discussion and this point is also discussed there (lines 289-302). Anyhow, the results will allow the generation of hypotheses that deeper study will be able to address. Comparison of a selected number of ORs from different species in the same lab and same set-up would indeed be a good step forward to address the differences between species, but we feel definitive conclusion would require the study of complete sets of ORs, available in the near future.

2. Fig 4a. Not enough information is provided either in the results section or figure legends or materials and methods to be able to understand this graph. Moreover, the idea that an odor receptor specificity increases as concentration of odorant drops has been shown before for fly and mosquito receptors. While I acknowledge that this piece of information is new for Slit odor receptors, there is nothing novel about this concept.

Answer

The result section (see lines 190-191) and the legend of the figure have been detailed for better understanding. We agree that the idea that an odor receptor specificity increases as concentration of odorant drops is not novel in insects and we just attend here to summarize the results of our dose-response experiments, which show that it is also true for Lepidoptera, as expected. We have changed the sentence, lines 204-206, as follows “our observations are in accordance with the current view that, at the peripheral level, combinatorial coding is likely to play a role in coding the variation of odorant quantity”.

3. Line 182: *“Taken together, our observations suggest that, at the peripheral level, combinatorial coding may not be the rule for coding odorant identity at low doses but is likely to play a role in coding the quantity of some odorants”.*

In a similar vein as point #2, the above observation maybe be a new finding for Slit Ors but is not really a significant finding with regards to olfactory information processing.

Answer

As explained above in our response to point 2, this sentence has been changed (lines 204-206).

4. Line 240: *Here, at low doses, we found 10 SlitORs narrowly tuned to widely occurring odorants, emitted by numerous flowering plants*

To claim that some receptors are very narrowly tuned and respond to only one or two odorants based on a very small panel (relative to other studies that have included much larger odor panels to test fly or mosquito odor receptors) is a reach.

Answer

We agree with the reviewer remark and we have changed this sentence as followed: “Here, at low doses, we found 10 SlitORs tuned to a limited number of odorants in our panel, all being widely occurring odorants, emitted by numerous flowering plants” (Line 263)

5. Line 189: *We therefore placed these functional data in a phylogenetic framework, in order to investigate the evolution of OR function in this insect order.*

While I think this is an interesting analysis and useful conclusion, I wish the authors are a little more explicit in describing the nature of functional data obtained from other studies. It must be noted that the functional data for odor receptors in other moth species were obtained using different methodologies. Although the authors take a more empirical approach for this analysis and while I believe the results are valid, I think caveats must be noted, either in the results section or in the Discussion section.

Answer

The different systems used to obtain the functional data reported here have been detailed in the figure 5 legend and we prefaced our statement in the Results part as follows: “.....,

although it must be noted that these ORs have been deorphanized using different expression systems and different odorant panels” (lines 213-214).

Minor concerns/suggestions;

1. First paragraph of discussion is usually reserved for the most important findings of the study. Placing the finding “Slit Ors are functional in the empty neuron” in this paragraph takes a bit away from the other important findings of the study, especially for a journal such as Nature Communications.

Answer

We agree and this paragraph has been moved to another part of the discussion where we now discuss limitation and caveats of our study (lines 289-302) as suggested in the point 2 just below.

2. Discussion section would benefit from a sub-section on any specific limitations and caveats regarding the conclusions of the current study

Answer

We agree and, as pointed in the answer to point 1, we have added such section lines lines 289-302.

Reviewer #2 (Remarks to the Author):

The manuscript by Fouchier et al represents a very worthwhile contribution to a better understanding of the molecular basis of olfaction. The authors have performed the most extensive functional analysis of a large array of OR candidates in a non-model organism so far. The authors show that the empty neuron technique is suitable for characterization of olfactory receptors outside Diptera. A basic conservation of function within the receptor repertoire of Lepidoptera, across the evolutionary radiation of different major clades is highlighted. With no doubt this study will serve as a reference for further studies in this field.

The study is very well designed, taking advantage of a multitude of relevant techniques/approaches. The results appear solid and are discussed in a balanced way, and the manuscript is well written with nice and informative figures.

My comments are all relatively minor:

Lines 47-48: “paves the way for the development of innovative olfactory-based control strategies” – this is not very specific and could just as well be deleted. Not clear (not even in the main text) what the significant contribution to future applications in pest control is. I think that the authors should stick to the molecular basis of olfaction and refrain from claiming applied implications.

Answer We agree and this sentence has been deleted.

Lines 50-52: First sentence could be deleted. Just idling! Second sentence is a good enough (actually better) start of the introduction.

Answer We agree and this sentence has been deleted.

Lines 93-106: I think that this paragraph could be improved to more clearly describe how the 47 OR candidates from the transcriptome were “sampled”, which were functionally characterized and “what about the rest”. This is partly dealt with in the discussion (lines 212-216) but I think that there is room for improvement.

- 47 candidates were identified in the transcriptome
- These candidates were distributed among 17 of the 21 highly supported clades in the phylogeny. Any comment on the fact that there were no Slit representatives in the other four clades?
- Then 35 (out of 47) of the candidates belonging to the 17 lineages were characterized. How were these 35 selected? Why were the other 12 discarded at this point?

Answer

Within the 47 ORs available, we have sampled those that were full length for functional studies (35 ORs) and the incomplete 12 other ORs were discarded. This is now better explained in the Results part (line 105). Unfortunately, although we tried to sample ORs in all the 21 lineages, some lineages did not contain full length ORs and were thus not represented. Anyhow, the 17 lineages that contained the studied full length ORs ensure a good cross-section of the OR diversity (lines 108-111).

- 5 of the 35 did NOT express in the *Drosophila* antenna? Any comment on why? Did this bias the results in any way?

Answer

Yes, 5 ORs did not express in *Drosophila* antennae (cf supp table 1). In the same way, similar issues with OR expression have been observed in previous studies (Hallem & Carlson 2006; Carey et al 2010). We have no explanation for that, as these unexpressed ORs are randomly scattered in the phylogeny and thus it may not be a question of sequence/conformation/function. It is thus difficult to comment on that in the text and to predict any ensuing bias in the results, although we cannot exclude any.

- 6 out of the 30 were considered “non-functional”. I would like to see a somewhat more extensive discussion of the criteria used and possibly some references to other studies (to help somebody not really familiar with the empty neuron system to understand what this “non-functionality” means).

Answer

ORs were qualified as non-functional when they were found to be expressed at the RNA level in *Drosophila* antennae (using RT-PCR) but the neurons supposed to express them were still “empty”, as reflected by the absence of regular spontaneous background neuronal activity. In supp fig 1a, we show a representative recording of an ab3a neuron from a fly line expressing a SlitOR categorized as non-functional. Legend has been modified to clarify that. Supp fig 1b and c show regular spontaneous firing activities. Neuron spiking in abnormal bursts have been previously characterized as “empty neurons” by Dobritsa et al 2003 Neuron (figure 5f in this reference). This reference has been added to the main text and details on the meaning of non-functional ORs has been added in the text (lines 115-118) as follows: “For 6 lines, ab3A neurons displayed an abnormal spontaneous firing rate, with bursts of action potentials (Supplementary Fig. 1a). This phenotype was reminiscent of that observed in mutant ab3A neurons that do not express any receptor (Dobritsa et al 2003) and indicated that these SlitOR transgenes, while expressed, were likely non-functional.”

Apart from that it is pleasing to find that the success rate was similar to what has previously been obtained in *A. gambiae*. Well done!

- Then four out of the 24 did not display any significant response to the stimuli. Isn't the most likely explanation that relevant ligands for these receptors were not included in the panel? The same may hold for the three that showed very low responses.

Answer Sure this is the most likely explanation. This has been added lines 130-131.

Line 123: I do not follow. Delete “only”?

Answer This word is indeed not relevant. It has been deleted.

Lines 148-152: I am a bit confused. The size of a bubble represents the global response of OR repertoires of the different species? But what determines the position of a bubble in the PCA

plot? Does each bubble represent a compound? Maybe I am just ignorant but then I may still not be the only one. Please provide a better explanation.

Answer

The reviewer can refer to our answer to first comment of reviewer #1 upper. In fact, each bubble represents an odorant. The 2D position of the bubble is determined by a PCA of 22 molecular descriptors of the odorant. The size of the bubble represents the response of the OR repertoire to the odorant. It is derived from the Euclidian distance from the odorant to the solvent. This type of plot has been used previously in Carey et al. 2010 Nature. We have now better explained what this figure represents in the text (see lines 165-170) and the Method part (see lines 384-389).

Line 160: It is surprising to me that no ORs responding to pheromone components were included. I would imagine that the responses were quite high, especially if corrected for the relatively low volatility of these compounds compared to many of the plant compounds.

Answer

The pheromone receptors SlitOR13 and 6 have been already studied in two previously published works (Montagné et al 2012, de Fouchier et al 2015). They were expressed in a different empty neuron system (at1) that is better suited for pheromone receptors (i.e that gives stronger responses to pheromone compounds than ab3). Furthermore, the method used here to compensate for odorant volatility (Andersson et al. 2012) is only suitable for the more volatile compounds formulated in paraffin oil, in contrast to the pheromone compounds (formulated in hexane) used here.

Lines 187-188: How were the species chosen to be included in the phylogeny? (see lines 360-363. No explanation for the selection). How robust is the topology of the tree if it is constructed with different species included? Please comment.

Answer

We have now explained in the Method section how the species were chosen for the phylogeny, see lines 401-403: “In order to take into account the taxonomic diversity of Lepidoptera, the OR phylogeny was built from OR sequences from 7 different lepidopteran families. For each family, the species in which the OR repertoire was best annotated was chosen.” Because of such sampling of species in representative lepidopteran families, the phylogeny is thus robust.

Lines 300-301: “equipped with a 50x magnification objective” but what about the total magnification? Ocular? 50x magnification appears low to me for this kind of work.

Answer We used 15x oculars that makes 750x total magnification. This has been clarified in the Method part line 338.

Lines 320-322: Why was the counting window shifted from 100 ms to 400 ms for the pheromone compounds? Does not make any sense to me. Requires explanation.

Answer Different time lags were used for pheromones and other compounds to take into account the time for the molecules to reach the antenna. After stimulation by pheromones, neuron response has been observed to start later than when stimulated with more volatile

compounds, because of their different volatility. This has been explained in the Method part, see line 358.

List of references: May need some attention. Several scientific names are not in italics.

Answer We have now corrected all references for species names.

Figure legend 1: Add info on what the letters in the circles represent.

Answer Letters in the circles point the 21 OR clades. This has been clarified in the figure legend (line 692-693)

Figure 2a: The authors write (Z,E)-9,11-14:OAc but for instance (Z)9-14:OAc. I think it should be (Z)-9-14:OAc to be consistent. This discrepancy holds also for other compounds (e.g. (E)2-hexenal) and for supplementary table 2 where I also think that there should be room to spell out the full names of the pheromone components together with the short forms.

Answer We have now adopted the correct nomenclature (e.g. (Z)-9-14:OAc) throughout the manuscript and in figures and supplementary information. In supplementary table 2, full names of pheromone components are now given together with the short forms.

Figure 3: Why are OR13 and OR6 not included? Doses for the pheromone compounds in Fig 2A are 10x lower than for the other compounds and based on this I consider the responses in Fig 2A significant (and should qualify the receptors to be compared with the others in Fig 3).

Answer Responses of OSNs expressing SlitOR6 and 13 to some pheromone compounds were indeed significant. Since these ORs have been studied in depth in previous publication (see Montagné et al 2012, de Fouchier *et al.* 2015) and as volatility of long-chained pheromone molecules cannot be compensated by the model of Anderson et al (2012) (as they are diluted in hexane, not paraffin oil), we did not include dose response curves for these ORs in this figure (see also response to comment above from this reviewer).

Supplementary figure 1: It is not clear to me why (a) but not (b) represents a non-functional receptor. Please explain the criteria. Also explain the difference between the cells (different spike amplitudes). It is the cell firing with a large spike amplitude that is relevant in this study?

Answer Yes, in supplementary figure 1a and b one should consider the large amplitude spikes only, as they correspond to the ab3A neuron that is targeted for OR expression. The large amplitude spike frequency is clearly different between supplementary fig a and b. One can see in a) abnormal bursts of spikes, reminiscent of what has been observed in an empty neuron (i.e. without any expressed OR, see Dobritsa *et al.* 2003 Neuron, and comments above on the reviewer remark about “non-functional” ORs). In b), one can see regular large amplitude spikes, typical of an OR-expressing neuron. This has been clarified in the legend of supplementary fig 1.

Supplementary figure 3: Improve the layout (introducing some space between the “pairs” to make it more clear what should be compared.

This has been done.

Reviewer #3 (Remarks to the Author):

Review of de Foucher et al.: “Functional evolution of Lepidoptera olfactory receptors revealed by deorphanization of a moth repertoire”

For: Nature Communications

This paper by de Foucher et al. is an important contribution to the field of insect olfaction, and to animal olfaction in general. The results are novel and should be of interest to readers interested not only animal olfaction, but also to those interested in neuroethology, evolution, and animal behavior. A crucial finding in this study is that the *Drosophila* “empty neuron” technique can be successfully used for non-dipteran investigations of OR functionality, providing a robust platform for the de-orphanization of non-dipteran ORs. The finding that a good many of the OR response profiles matched nearly perfectly the response profiles of known, native *Spodoptera littoralis* olfactory sensory neurons (OSNs) is significant in that it further validates the novel use of the empty neuron technique in this study: i.e., the lepidopteran ORs are shown to behave normally in this dipteran expression system. The paper provides a first-ever basis for comparing odorant receptor (OR) function across the Lepidoptera for so-called “general odorants” comprising the vast majority of molecules encountered by insects in various environments. Moreover, the results strongly suggest a link between the importance of certain odorants with regard to the life histories of moth and fly species according to the narrowness of OR tuning profiles in response to these ligands. To their credit, the authors use dose-response profiles and investigate responses to very low emitted amounts of odorants, thereby demonstrating the true narrowness of ligand specificity of a large number of ORs. The authors gamely construct a phylogeny of lepidopteran OR-ligand couplings along with the chemical classes of these ligands to then relate these patterns to the ORs that might be more recently evolved.

I have some minor editorial corrections/suggestions throughout the manuscript, plus two more major comments:

1) I wonder if there was never an inhibitory response to any of the odorants? I assume there wasn't because the profiles in Figure 2b do not show any negative values and the method for subtracting out the background firing rate from the post-stimulus-delivery firing rate would have revealed a sub-background level of firing that would have been displayed in Figure 2b. If this is so, I feel it would be helpful for the authors to state this outright somewhere, i.e., that there was never a reduction in background firing rate to any of the odorants tested. The narrow-vs.broad-tuning OR profiles in the similar histograms of the Hallem and Carlson *Drosophila* work nearly all exhibited a good proportion of negative values (inhibition) in response to some of the >100 odorants they tested. Their results may or may not represent a significant difference between lepidopteran ORs and *Drosophila* ORs' tuning characteristics, but maybe it does?

Answer

We never observed an inhibitory response of the SlitORs when expressed in *Drosophila* antennae. Accordingly, inhibitory response has never been observed for *Spodoptera littoralis* OSNs (Binyameen *et al.* 2012 and 2014) nor, to our knowledge, for other Lepidoptera (e.g. Ghaninia *et al.* 2014). Since this is a different situation than what has been observed in *Drosophila*, we agree it deserves comments that have been added lines 127-129.

2) On lines 183 and 184, I think the statement that “combinatorial coding may not be the rule for coding odorant identity at low doses....” is not entirely correct. Combinatorial coding would be used even if only one glomerulus is activated by one specifically tuned type of OR/OSN because the activity of the neuropil in that one glomerulus is nevertheless being compared against the total inactivity of the other glomeruli. If activity in other glomeruli begins to occur at higher doses, the relative intensities of glomerular activity would still be centered around the higher activity in that first glomerulus. This is just part of a continuum of combinatorial coding of odor quality at very low vs. higher odorant intensities.

Answer We agree and this point has been also noticed by reviewer 1. As explained in response to referee 1, the sentence has been rewritten (see lines 204-206).

Minor comments, most having to do with readability:

Line 96: might be better to say, “In this tree, ORs fell into groups comprising 21 highly..”. (To me, this sentence at first did not appear to have a verb; it took careful re-reading to realize that “clustered” was in fact the verb).

done

Line 151: may be better to say: “Whereas this presentation is at this point limited...”

done

Lines 152-153: I would suggest: “...only in mosquito and fewer aliphatics have been tested on SlitORs), we felt this analysis might nevertheless reveal interesting features.”

done

Line 153: use “Although” instead of “While”

done

Line 154: use “effective” instead of “efficient”

done

Line 178: replace “on” with “in eliciting”

done

Line 195: replace “grouped” with “fell”

We changed the sentence as followed “Clades L to U formed a monophyletic group” (line 219)

Line 209: use “because” rather than “as”

This sentence has been removed following reviewer #1 suggestion (first minor comment, see above)

Line 211: replace “considered” with “compared”

This sentence has been removed following reviewer #1 suggestion (first minor comment, see above)

Line 212: Wouldn't "functionally characterized" be more precise than "functionally expressed"?

"functionally expressed" have been modified to "some ORs were not functional"

Line 214: replace "and" with "as well as"

done

Line 216: I'm not sure of the technical terminology here, but would suggest that "inactive" might be a better word for better readability than "refractory".

Refractory has been changed to "non-functional"

Line 220: better to say, "...appeared particularly effective at responding to..."

We think the meaning may be changed with these words. We thus propose to write this sentence as following: "SlitORs appeared particularly effective at distinguishing short-chain aliphatic alcohols ..."

Line 222: replace "one" with "repertoire" (even though "repertoire" will be used twice in this sentence)

done

Line 224: change to, "...plant odours emitted by both flowers and leaves."

done

Line 229: replace "whereas" with "and"

done

Line 230: remove comma

done

Line 238: replace "While" with "Although"

done

Line 239: better to say, "...herbivorous insects discriminate host from non-host..."

done

Lines 240 and 241: remove commas

done

Line 247: replace "brought" with "provided"

done

Line 250: add "but", i.e., "the same odorant but with different detection thresholds..."

done

Line 251: replace "detect" with "characterize"

done

Line 272: replace "appear as" with "appear to be"

done

REVIEWERS' COMMENTS:

Reviewer #1 (Remarks to the Author):

The authors have satisfactorily addressed every one of my concerns with the previous version of the ms. I have no further concerns with the revised version of the ms.

Reviewer #2 (Remarks to the Author):

The authors have responded in a satisfactory way to by comments and made appropriate changes in the revised version of the manuscript.